# Understanding Viral Infection Mechanisms and Patient Symptoms for the Development of COVID-19 Therapeutics

**DOI:** 10.3390/ijms22041737

**Published:** 2021-02-09

**Authors:** Hyung Muk Choi, Soo Youn Moon, Hyung In Yang, Kyoung Soo Kim

**Affiliations:** 1Department of Clinical Pharmacology and Therapeutics, Kyung Hee University School of Medicine, Seoul 02447, Korea; chl2813@khu.ac.kr; 2Division of Infectious Diseases, Department of Internal Medicine, Kyung Hee University Hospital at Gangdong, Gandong-gu, Seoul 02447, Korea; sooyounmoon78@gmail.com; 3East-West Bone & Joint Disease Research Institute, Kyung Hee University Hospital at Gangdong, Gandong-gu, Seoul 02447, Korea; yhira@khu.ac.kr

**Keywords:** SARS-CoV-2, COVID-19, ACE2, TMPRSS2, camostat mesilate, immunomodulation

## Abstract

Coronavirus disease 2019 (COVID-19), caused by the SARS-CoV-2 virus, has become a worldwide pandemic. Symptoms range from mild fever to cough, fatigue, severe pneumonia, acute respiratory distress syndrome (ARDS), and organ failure, with a mortality rate of 2.2%. However, there are no licensed drugs or definitive treatment strategies for patients with severe COVID-19. Only antiviral or anti-inflammatory drugs are used as symptomatic treatments based on clinician experience. Basic medical researchers are also trying to develop COVID-19 therapeutics. However, there is limited systematic information about the pathogenesis of COVID-19 symptoms that cause tissue damage or death and the mechanisms by which the virus infects and replicates in cells. Here, we introduce recent knowledge of time course changes in viral titers, delayed virus clearance, and persistent systemic inflammation in patients with severe COVID-19. Based on the concept of drug reposition, we review which antiviral or anti-inflammatory drugs can effectively treat COVID-19 patients based on progressive symptoms and the mechanisms inhibiting virus infection and replication.

## 1. Introduction

The outbreak of a novel coronavirus was reported in Wuhan, in the Hubei province of China, in December 2019. This virus was officially designated as severe acute respiratory syndrome coronavirus 2 (SARS-CoV-2) because of its phylogenetic and taxonomic similarities to other coronaviruses. This virus is generally called COVID-19 [1]. After its outbreak in December 2019, the World Health Organization (WHO) declared that the COVID-19 outbreak was a pandemic in March 2020. As of the end of December 2020, about 90 million cases have been confirmed in over 219 countries. Its symptoms range from mild fever to cough, fatigue, severe pneumonia, acute respiratory distress syndrome (ARDS), and organ failure. The mortality appears to be around 2.2% worldwide, though it varies according to patient age, disease severity, and other circumstances, including patient comorbidities [2]. However, there are no licensed drugs or definitive treatment strategies for patients with severe COVID-19. Only antiviral or anti-inflammatory drugs are used as symptomatic treatments based on the experience of each clinician.

This pandemic, with its high fatality and transmission rates, has completely changed our daily lives, but it is all the more confusing, because SARS-CoV-2 vaccination schemes aimed at achieving herd immunity are being delayed and we do not know when the pandemic will end [3]. Therefore, it is necessary to find effective drug and treatment strategies for clinicians as soon as possible. However, it will take a long time to develop a completely new drug. Thus, based on the concept of drug reposition, in which existing drugs are applied to other diseases at low cost and in a shorter time, it is necessary to discover which of the drugs currently in use can help treat COVID-19 patients [1,4]. Repositioned drugs are being discovered through specific pharmacological insights or experimental screening platforms [5,6]. Conventional antiviral or anti-inflammatory drugs have the potential to be developed as treatments for COVID-19. However, there are several challenges with drug repositioning. Establishing biochemical, pharmacological, and clinical evidence is difficult due to the lack of qualified data. In addition, it is also necessary to determine the effective drug concentration and to devise a method to distribute the drug to the target organs. Basic medical scientists, who try to develop therapeutic agents against COVID-19, also need to understand the more recent clinical characteristics of COVID-19 and the basic molecular characteristics of this coronavirus. For this, we need to know the information about COVID-19 that is available to basic scientists. Here, we review several questions: What is SARS-CoV-2? What are the mechanisms by which cells are infected, the clinical features of COVID-19, and the mechanisms through which COVID-19 can be effectively treated with existing drugs? Answers to these questions will help to provide new insights to aid the development of therapeutics in light of the various clinical and basic research studies currently underway.

## 2. Understanding of SARS-CoV-2 and COVID-19

### 2.1. Classification of SARS-CoV-2

SARS-CoV-2 causes COVID-19, and thus is named the COVID-19 virus. This virus is classified among beta-coronavirus family members along with MERS-CoVs and SARS-CoVs [7]. A study has reported that SARS-CoV-2 has more than 79% nucleotide identity with that of SARS-CoV [8]. This coronavirus, which is distinguished by a transmembrane spike protein (S protein) with a crown-like appearance, is a positive-sense, single-stranded RNA virus (+ssRNA) that is approximately 60~140 nm in diameter. The Coronaviridae family is divided into alpha, beta, delta, and gamma coronaviruses (CoV) taxonomically. Among them, only alpha and beta coronaviruses are infective to humans. Until now, seven human coronaviruses (HCoV) have been identified, including alpha HCoV-NL63 and 229E, beta HCoV-OC43, HKU1, SARS-CoV, MERS-CoV, and SARS-CoV-2 [9].

### 2.2. Roles of SARS-CoV-2 Structural Proteins in Infection

The genome of SARS-CoV-2 consists of 29,891 base pairs of nucleotides and 9860 amino acids, which encode spike (S), nucleocapsid (N), membrane (M) and envelope (E) proteins (Figure 1). In particular, electron microscopy images of the spike protein show a spherical surface with crown-shaped spikes measuring 9–12 nm [10]. The SARS-CoV-2 spike protein has two subunits; S1 induces viral attachment to host cell surface receptors and S2 mediates the fusion of host cells with viral membranes. The S1 N-terminal and C-terminal domains interact with host receptors. The C-terminal domain includes a receptor-binding domain (RBD). Thus, the RBD receptor-binding motif is attracting attention as a determinant of host affinity, and several mutations to the spike protein have been reported [10]. In particular, the mutation of the spike protein at position 614 (aspartic acid to glycine, D614G) is drawing attention as the variant containing G614 becomes the predominant form worldwide. Because G614 is more infectious than D614, it was assumed to spread rapidly [11]. Under this hypothesis, studies show that the G614 variant tends to have higher viral RNA levels and pseudovirus titer than the D614 variant in in vitro experiments [11,12]. More recent studies have shown that the G614 mutation has a higher viral load in the upper respiratory tract than the D614 mutation in animal models and human cells, but increases susceptibility to neutralizing antibodies [13,14]. In addition, mutations in spike proteins have been identified at various locations, such as A222V, S477N, N501, H69-, N439K, Y453F, S98F, D80Y, A626S, and V1122L. In particular, S477N, N501, H69-, and N439K are mutations of the receptor-binding domain (RBD), which could lead to a change of the recognition site of the antibody and an increase of its binding affinity to angiotensin-converting enzyme 2 (ACE2). This could result in increased pathogenic potential. More research should be done to clarify the above (https://nextstrain.org/ncov/global).

Additionally, the N protein encloses the viral genome as it binds to coronavirus RNA and forms a capsid. The N protein seems to be involved in viral assembly, transcription, and budding. In particular, the RNA-binding domain of the N protein modulates its interaction with RNA and cellular signaling with the host cell [15]. Thus, the RNA-binding domain inhibits the production of cytokines by interrupting signaling pathways associated with interferon and RNA interference in host cells [16,17]. The M protein, which is located between the membrane and capsid, determines viral shape as it binds to the nucleocapsid. Additionally, the M protein is dominantly localized in Golgi and trafficking vesicles during coronavirus assembly. This implies that the M protein is essential for SARS-CoV-2 maturation [18]. The E protein is the smallest structural protein, which acts as a viral ion channel. Although it seems to take part in virus assembly, budding, and envelope formation, the specific role of the E protein is largely unknown [19].

### 2.3. SARS-CoV-2 Life Cycle and Its Infected Organs 

SARS-CoV-2 is usually transmitted by inhalation or contact with infected droplets. Inhaled droplets or aerosol carrying the virus then infect and spread through the respiratory tracts. The virus incubation period is roughly 2 to 14 days (median 5.2 days). During this time, virus particles are present in the secretions of infected persons. It has also been confirmed that asymptomatic patients can transmit the virus after the virus incubation period among other persons [1]. The WHO has estimated the virus reproduction number at 1.4 to 2.5, but other studies assumed it to be around 3.3 during the early spread of COVID-19 [20,21]. Additionally, viral transmission could be possible for up to 10 days after the first symptom onset for moderate or mild cases. Ten to twenty days after first symptom onset, replication-competent virus has been found in severe patients, but not in mild or moderate patients [22]. First, viral replication seems to take place in the mucosal epithelium of the upper respiratory tract. Then, further multiplications occur in the lower respiratory tract and gastrointestinal mucosa. At this stage, most patients show mild viremia or respiratory symptoms [23]. Additionally, others patients have shown non-respiratory symptoms, including headache, diarrhea, and conjunctivitis [2]. This multi-organ involvement of COVID-19 may be attributable to the universal expression of ACE2 receptors, which the virus uses to enter cells [24]. Thus, the sustained viral replication and subsequent immunoreaction in the above organs has also resulted in lung, heart, and kidney injury, as well as pneumonia [25].

### 2.4. Role of the Spike Protein and ACE2 in SARS-CoV-2 Entry

#### 2.4.1. Cleavage of Spike Protein for Its Attachment to ACE2

The SARS-CoV-2 spike protein is a complex composed of S2 and S1 subunits, which work as transmembrane proteins and bind to ACE2, which is the host cell virus receptor (Figure 2) [26]. The human ACE2 is a functional receptor that is mandatory for the cellular entry of SARS-CoV-2 [27]. Cellular serine protease TMPRSS2 is employed to prime the spike protein [28]. Additionally, furin, which is a kind of serine protease dominantly distributed in pulmonary and intestinal tissues, activates only the SARS-CoV-2 spike protein, but not that of SARS-CoV. This could explain why SARS-CoV-2 has a 10~20 times higher affinity for ACE2 receptors compared to SARS-CoV [29]. A recent study indicated that modulation of ACE2 glycosylation could control the binding affinity of the spike protein with its receptor. For example, glycosylation on N90 of the S1 subunit interrupts its binding to ACE2, but glycosylation on N322 stimulates binding to ACE2 [30].

As the ACE2 receptor is widely expressed in tissues, i.e., the nasal mucosa, bronchus, lungs, stomach, heart, kidneys, and ileum, these organs are vulnerable to COVID-19 virus infection. Thus, it is assumed that symptoms of COVID-19, such as respiratory symptoms, heart injury, kidney failure, diarrhea, and vomiting, may imply involvement of multiple organs [25]. COVID-19 could result in ARDS through interruption of cell signaling pathways in host cells [31]. Binding of SARS-CoV-2 to ACE2 downregulates ACE2 expression. In vitro treatment with recombinant SARS-CoV-2 spike protein could also downregulate ACE2 expression by binding to ACE2 and inhibiting viral attachment to ACE2 [31]. 

#### 2.4.2. Role of ACE2 in the Renin-Angiotensin System (RAS) 

The ACE2 receptor plays an important role in the RAS, which regulates blood volume, vascular resistance, and inflammatory reactions. As shown in Figure 3, ACE, which is carboxypeptidase, activates angiotensin I, then angiotensin I activates vasoconstrictor angiotensin II, which degrades vasodilator bradykinin, resulting in an increase in blood pressure. On the contrary, ACE2 counteracts ACE function by degrading angiotensin I and angiotensin II into angiotensin 1-7, which are potent vasodilators, leading to a decrease in blood pressure. Additionally, it is well known that angiotensin 1-7 and ACE2 alleviate inflammation and resist oxidative stress. Furthermore, activation of ACE2 receptors reduces inflammation levels and delays lung, kidney, and heart fibrosis [27,32,33]. Likewise, an experiment showed that SARS-CoV infection significantly increased the expression of angiotensin II and decreased ACE2 expression in mice [34]. Another study indicated that rats with ARDS have decreased expression of ACE2, but an increased expression of angiotensin II [35]. From the above results, it can be inferred that COVID-19 virus-induced RAS imbalance causes ARDS.

Several observational studies have reported that diseases such as hypertension, cardiovascular diseases, and diabetes mellitus are more common in COVID-19 patients than healthy people [36,37,38]. Additionally, COVID-19 patients with hypertension, cardiovascular diseases, and diabetes mellitus have higher mortality rates than people with no comorbidities [39]. Those diseases are the most common comorbidities and lead to worse prognosis. Although more research is needed, the reason for the high mortality of COVID-19 patients with vascular diseases could be due to two causes: aggravated vascular dysfunction with SARS-CoV-2 infection and differing baseline statuses of patients, including confounding factors. It is well known that SARS-CoV-2 infection induces a hypercoagulable state by over-activating inflammatory mediators and downregulating ACE2 [40,41]. Thus, SARS-CoV-2-induced hypercoagulable state may aggravate cardiovascular disease and diabetes mellitus, leading to fatal thrombotic events [42]. However, COVID-19 patients with hypertension or diabetes mellitus also tend to have baseline risk factors, namely age >65 and respiratory and vascular diseases. In addition, patients with hypertension usually take ACE inhibitors (ACEi) or angiotensin II receptor blockers (ARBs), which could work as a confounding factor [39]. It is still unknown whether taking ACEi or ARB is helpful, neutral, or harmful for COVID-19 infection and progression [43]. These studies show that SARS-CoV-2 infection through ACE2 host cell receptors could cause vascular damage, as well as respiratory diseases, but more studies need to be conducted to find out the two-sided role of ACE2 in COVID-19 infection.

## 3. COVID-19 Pathogenesis through Inflammation-Mediated Tissue Damage

### 3.1. SARS-CoV-2-Induced Inflammation and Cytokine Storm

Severe inflammatory response is a remarkable feature of COVID-19 symptoms. Clinically, patients with COVID-19 have higher serum levels of proinflammatory markers that include IL-1β, IFN-γ, IP-10, MCP-1, IL-4, and IL-10. In addition, patients in intensive care units (ICUs) have higher serum levels of IL-2, IL-7, IL-10, GCSF, IP-10, MCP-1, MIP-1A, and TNF-α than non-ICU patients [2]. Similarly, a meta-analysis of 2984 cases from 18 studies reported that serum levels of IL-6, IL-10, and ferritin were much higher in patients with fatal COVID-19 disease than in patients with mild symptoms [44]. The deadly inflammatory reaction in severe patients can be explained by 3 factors: viral replication causes cellular impairment, decreased expression of ACE2 receptors, and overactivated immune cells [45]. COVID-19 dominantly infects lung epithelial cells, alveolar macrophages, nasal goblet cells, vascular endothelia, and ileal cells. As these cells have a high expression level of ACE2 receptors, it can be inferred that SARS-CoV-2 could infect various tissues, as well as lung cell fields [46,47]. In particular, SARS-CoV-2 infection of the respiratory system and its subsequent viral replication cause pyroptosis, which rapidly removes intracellular pathogens through a proinflammatory programmed cell death. This can cause vascular leakage and the massive release of proinflammatory cytokines such as IL-1β, IL-6, and IFN-γ, resulting in apoptosis of infected cells [48,49]. Furthermore, the serum level of damage-associated molecular patterns (DAMPs), such as ATP, uric acid, and DNA; pathogen-associated molecular patterns (PAMPs), such as viral RNA; and lipoproteins increase as a result of pyroptosis and its subsequent inflammatory reactions [50]. Then, alveolar epithelial cells and macrophages sense these molecular patterns and accelerate the release of inflammatory cytokines, which include IL-6, IFN-γ, and MCP1 [51,52]. These cytokines induce monocytes and T lymphocytes to infiltrate into infected sites from blood serum and also lead to apoptosis of infected cells [44]. In support of this scenario, about 80% of COVID-19 patients show blood lymphopenia, which implies enormous T cell infiltration, cellular pyroptosis, and inflammation [45,49].

As mentioned in Section 2.4, SARS-CoV-2 infection downregulates ACE2 receptor expression and aggravates inflammation. It is well-known that the activation of ACE2 receptors mitigates inflammation and fibrosis by inhibiting the ACE–angiotensin II–angiotensin I type 1 receptor axis [32]. The apical surface of the airway epithelium has a high level of ACE2 expression. Thus, downregulation of ACE2 and failure to inactivate the angiotensin II–angiotensin I type 1 receptor pathway causes chronic inflammation that leads to ALI or ARDS [33,53]. Other studies even suggested that ACE2 may play a pivotal role in protecting cardiovascular and intestinal health by mitigating inflammation [54,55]. In other words, SARS-CoV-2 infection induces ACE2 downregulation and worsens inflammation. Additionally, we could infer that uncommon symptoms of COVID-19, including gastrointestinal symptoms (diarrhea and vomiting) and vascular and pulmonary diseases, could result from interrupted RAS function and inflammation. Activation and differentiation of T cells accelerate and amplify immune responses. The activated T cells further produce proinflammatory cytokines and recruit more immune cells, such as lymphocytes and leukocytes, into inflammatory sites [56]. The proinflammatory cytokines above are indicators of T helper 1 cell activation and lymphocyte recruitment [50]. In most clinical cases, those recruited cells temporarily increase inflammatory reactions but finally kill the infected cells. The production of antibodies against a virus in B cells is promoted by helper T cells, and virus-infected cells are cleared by cytotoxic T cells. T-cell-induced inflammatory reactions help recruit lymphocytes and leukocytes into infected tissues [56]. This implies that initial inflammatory reactions are essential to cope with viruses. In most patients, after the virus and infected cells were removed, inflammation was relieved.

However, a relatively small portion of COVID-19 patients show severe inflammation. This is caused by delayed viral clearance, which induces chronic systemic inflammation and widespread tissue damage, even leading to cytokine storms [57]. In particular, severe COVID-19 patients show highly activated but decreased peripheral CD4 and CD8 T cells. The CD8 T cells have a high concentration of cytotoxic granules and most CD4 T cells are proinflammatory CCR6+ Th17 subtypes of CD4 T cells [58]. Thus, this explains the reason why COIVD-19 patients have immune dysfunction and delayed viral clearance leading to cytokine storms. In particular, the cytokine storm is considered to be one of the major causes of ARDS and the multiple organ failures that are seen in severe or fatal COVID-19 patients [59]. Thus, the downregulation of proinflammatory reactions has to be one target of urgent clinical investigations into new medicines that can successfully treat severely ill patients [60].

### 3.2. Symptoms Seen in COVID-19 Patients 

The optimal growth temperature range for most human coronaviruses is 33 to 35 °C, which infect the upper respiratory tract but do not cause systemic disease [61]. However, animal coronaviruses, including SARS-CoV and MERS-CoV, actively replicate at 37 °C, which means that they can infect the lower respiratory tract and even cause systemic disease. In addition, SARS-CoV and MERS-CoV are known to damage lung, kidney, liver, and gastrointestinal tissues, and even deplete immune cells [61,62]. Likewise, symptoms of COVID-19 vary from non-specific symptoms such as fever, cough, fatigue, diarrhea. and vomiting; to serious symptoms such as dyspnea, hypoxia, multiorgan failure, and disseminated intravascular coagulation [63]. A meta-analysis that included 1994 COVID-19 patients from 10 studies reported that symptoms included fever (88.5%), cough (68.6%), fatigue (35.8%), dizziness (12.1%), diarrhea (4.8%), and vomiting (3.9%) [52], and it seems that COVID-19 patients commonly experienced symptoms in the order of fever, cough, nausea or vomiting, and then diarrhea [64]. Other studies reported arthralgia, chest pain, rash, palpitation, and more seriously, ventricular dysfunction, pulmonary failure, acute kidney injury, and coagulopathy as complications of COVID-19. Additionally, it is noteworthy that dermatologic (rash, alopecia), neurological (olfactory dysfunction, sleep dysregulation, and memory impairment), and psychiatric (depression, sudden mood swings) symptoms have been reported among several COVID-19 patients [65,66]. Interestingly, the severity of COVID-19 symptoms varies among patients. About 20% of diagnosed patients were asymptomatic and 60% of them had mild symptoms. Only 20% of patients were hospitalized due to difficulty breathing [67,68]. Similarly, a meta-analysis in April 2019 that included 47,344 patients from 21 clinical studies showed that the risk of severity including ARDS and acute cardiac injury (ACI) was 18.0%. ARDS and ACI were associated with a 3.2% mortality rate [36]. More recent studies report that about 10% to 15% of patients progress to severe or critical conditions and that the mortality is about 2% [69,70]. In particular, COVID-19 symptoms were usually more severe in older patients (> 60 years) who had lung disease, heart disease, or hypertension comorbidities. This is relevant to weakened immunity, underlying respiratory diseases, or the expression level of viral receptors in the host’s cells [71].

The incubation period, which means the interval between infection and the onset of first symptoms, is generally 5~6 days, even though this can range from one day to as much as two weeks. Additionally, peak viral titer coincides with the time when symptoms first appear [72]. Most patients, who have mild or moderate symptoms such as fever, fatigue, and cough, recover in two weeks. However, 15% to 20% of patients suffer from difficulty in breathing and have high levels of inflammation with delayed viral clearance. Sustained inflammation induces further tissue damage, which leads to pneumonia, ARDS, ALI, and worsened prognosis [73].

### 3.3. COVID-19-Induced Tissue Damage and Clinical Presentations 

Interstitial pneumonia and subsequent ARDS are the leading causes of death in patients with COVID-19. SARS-CoV-2 damages lung tissues by infecting type 2 alveolar cells and lung endothelial cells [74]. Rapid viral replication can lead to virus-mediated ACE2 downregulation and uncontrolled inflammation. As a result, fibrinous exudates accumulate in the alveolar space, lymphocytic immune cells infiltrate into the alveolar space, and epithelial cells peel away. All of this interferes with oxygen exchange, causing ARDS [75].

As the arterial and venous endothelium expresses ACE2, the virus infects endothelial cells and disrupts the endothelial membrane. Viral particles are found in the endothelial cells of kidneys and lungs, partially explaining the lung and kidney failure of severe COVID-19 patients [74]. Additionally, hypercoagulable state is a distinctive characteristic of COVID-19 patients. SARS-CoV-2 infection induces endothelial disruption and endotheliitis. Endothelial disruption accelerates the secretion of von Willebrand factor, promoting the cross-linking between the surfaces of subendothelial cells and platelets. Endotheliitis stimulates the accumulation of macrophage and neutrophils in vascular bed. Additionally, COVID-19 induces RAS imbalance (mentioned in Section 2.4.2). For the above reasons, COVID-19 patients may overproduce thrombin and inhibit fibrinolysis, leading to micro thrombosis [76,77].

Myocardial injury, especially ventricular dysfunction, is reported in COVID-19 patients. ACE2 expression is high in cardiomyocytes, which allows direct viral injury. The virus has been isolated from myocardial tissue in several autopsy cases and relationships between viral load and myocarditis have been proven [78,79]. In addition, endothelial viral infection and subsequent inflammation are considered possible factors of myocardial infarction [76]. In particular, patients with existing vascular disease tend to highly express ACE2 in the endothelium, which explains why they have higher risk [80]. Furthermore, COVID-19-induced ARDS and pulmonary thromboembolism cause right ventricular failure or indirectly damage cardiomyocytes [81].

SARS-CoV and MERS-CoV are known to be neuro-invasive in human and are capable of both trans-neuronal and hematogenous routes [82]. Likewise, COVID-19 patients show neurological symptoms, including anosmia, neuroinflammation, and stroke. Although it is largely unknown, SARS-CoV-2 seems to invade the brain by retrograde axonal transport via the cribriform plate or systemic circulatory system [83,84] and damage neural tissues. However, some studies have inferred that nerve damage is due to inflammation and thrombosis caused by SARS-CoV-2 [77,83]. In addition, microvascular damage was found in brain samples from patients who died from COVID-19 without evidence of viral infection. The injury is not due to a lack of oxygen, but is usually related to a neuroinflammatory disorder [85].

## 4. Treatment Strategy According to COVID-19 Symptoms

### 4.1. Differencse in Viral Load between Mild and Severe COVID-19 Patients

When comparing 30 severely ill patients with 46 mildly ill COVID-19 patients, the average viral load in severely ill patients was 60 times higher, and virus clearance was also delayed more than in mildly ill patients (Figure 4). In particular, at the time of viral infection, delta values for cycle thresholds in real-time PCR for severe cases were much lower than those seen in mild cases. This means that severe cases have a drastic increase in viral load in the early disease stage and prolonged clearance times [86]. An observational study of 23 COVID-19 patients also found that their viral load increased rapidly at the same time as the onset of symptoms and then gradually decreased. Furthermore, the peak of the viral load was correlated with patient age. Viral loads of severe patients were about 1 log_10_ higher than those of mild patients, but the difference was not statistically significant [87]. Similarly, a study of two moderate COVID-19 patients revealed that viral loads peaked at 3 to 5 days from first symptom onset and slowly decreased until the second week. Additionally, virus was detected in their lower respiratory systems before lower respiratory symptoms occurred [88]. More recent studies showed similar results. A study on 1145 COVID-19 patients reported that viral load of patients and mortality is closely related (7% increase of hazard for each log copy per mL increase, hazard ratio = 1.07, *p* = 0.0014) [89]. The above studies commonly show that viral load drastically increases in the early infection period and peak viral load is closely related with patient prognosis.

### 4.2. Different Treatment Strategies for Different COVID-19 Symptoms 

Severe COVID-19 patients show different symptom changes over time compared to mildly ill patients (Figure 5). The levels of inflammatory factors, which include sIL-2R, TNF-α, high-sensitivity C-reactive protein (hs-CRP), and lactate dehydrogenase, decreased in mild patients until 10 days after the onset of symptoms. In contrast, the levels in severely ill patients rebounded 10 days after symptoms began. In addition, levels of inflammatory cytokines such as IL-6 and IL-8 continued to rise in critical patients [90]. In particular, severe patients showed pneumonia and widespread inflammation in the second week, which led to systemic inflammation, ARDS, cytokine storms, and multiorgan failure [91]. In contrast, patients with asymptomatic or moderate symptoms recovered from COVID-19 in the first week. This suggests that severely ill patients could not remove the virus properly, resulting in a severe inflammatory response. Additionally, persistent systemic inflammation causes ARDS, pulmonary fibrosis, hypoxia, and multiorgan failure. Thus, in severely ill patients, the inflammatory response 2 weeks after infection must be suppressed to prevent worsening of COVID-19 symptoms.

Meanwhile, mild symptoms such as fever, gastritis, and lymphopenia reflect early infection and response to viral load. Therefore, it makes sense to use an antiviral therapy, such as remdesivir or serine protease inhibitors, which block viral replication or viral entry into host cells during the initial onset of symptoms. Antiviral therapy in the early stages of infection significantly reduces viral load, shortens recovery time, and relieves respiratory symptoms. However, immunosuppression is not recommended because it can cause explosive viral growth. Furthermore, severe COVID-19 patients could suffer from pneumonia, hypoxia, and dysregulated immune reactions at 5~8 days from the onset of mild symptoms. Then, second week after symptoms begin, the serum levels of inflammatory markers such as CRP, IL-6, IL-8, and lactate dehydrogenase increase significantly, followed by fatal clinical symptoms such as ARDS, ALI, organ failure, and sepsis [91,92,93]. Therefore, for severely ill patients, strong anti-inflammatory interventions are recommended in the second week to prevent systemic inflammatory-mediated tissue damage.

## 5. Current Therapeutics for Treatment of COVID-19

The major mortality seen in COVID-19 patients may be attributed to ARDS and ALI caused by severely persistent inflammation [94]. Thus, antiviral therapeutics and anti-inflammatory agents are the best options for treatment of COVID-19 currently. Current therapeutics or medical options for COVID-19 largely depend on previous therapies used against SARS-CoV, MERS-CoV, influenza, and Ebola, because they have a common pathogenesis and genetic features. However, remdesivir is the only Food and Drug Administration (FDA)-approved therapeutic for COVID-19 patients, although about 7500 clinical studies have been registered on the WHO international clinical trials registry platform [95]. Currently, the following drugs are expected to treat COVID-19 due to plausible modes of action or unknown targets, including neutralizing antibodies that target spike glycoproteins, which are involved in host cell adhesion [2]; several antiviral and other drugs (e.g., hydroxychloroquine); 3CL protein inhibitors (ribavirin, lopinavir or ritonavir); RNA synthesis inhibitors (remdesivir, tenofovir disoproxil fumarate, and 3TC); neuraminidase inhibitors (oseltamivir and peramivir); and other small-molecule drugs (ACE2 inhibitors) [96]. Among them, an effective vaccine would provide the best prevention of COVID-19 spread, but it will take at least 6 months for the vaccination and formation of herd immunity [97]. Thus, current antiviral and anti-inflammatory agents are being tested to treat COVID-19 as a drug reposition strategy. Remdesivir and hydroxychloroquine have been the most tested drugs in clinical trials, but only remdesivir is recommended for antiviral treatment of COVID-19 patients. Additionally, dexamethasone is the most common therapeutic for COVID-19 patients who have severe inflammation. Globally, we are trying to find potential antiviral compounds based on the concept of drug reposition [1,25]. Remdesivir for Ebola, lopinavir or ritonavir for AIDS, favipiravir for influenza, ribavirin for hepatitis, and chloroquine or hydroxychloroquine for malaria have been tried. Among them, only remdesivir shows therapeutic efficacy, while lopinavir, favipiravir, and ribavirin do not [43].

### 5.1. Remdesivir and Hydroxychloroquine as Anti-Viral Drugs

#### 5.1.1. Remdesivir

As shown in Figure 6, remdesivir, which is a prodrug of adenosine analogue, inhibits the replication of RNA viruses such as SARS-CoV and MERS by blocking the enzymatic activity of RNA-dependent RNA polymerase (RdRP). Remdesivir showed antiviral effects in in vitro and in vivo experiments using rhesus macaques, resulting in lowered viral level and relieved lung damage [98]. Additionally, a randomized, double-blind, controlled trial (RCT) of 237 COVID-19 patients (158 in the remdesivir and 79 in the placebo group) in China reported no clinical benefit to the treatment of patients with remdesivir. Median values for clinical improvement were 21 days in the remdesivir group and 23 days in the placebo group, which were not statistically significant. The 28-day mortality in the remdesivir group was 14%, but that of the placebo group was 13% [99]. However, more recent studies showed different results. The RCT of 1063 patients (541 randomized to remdesivir and 521 to the placebo arm) showed superiority of remdesivir treatment over placebo effects in patients with severe COVID-19 [100]. The median discharge time for mild or moderate patients was 5 days in both the remdesivir and placebo groups. However, the times for severely ill patients were 11 days in the remdesivir group and 18 days in the placebo group. Additionally, the 29-day mortality was 11% in the remdesivir group and 15% in the placebo group [100]. In 397 hospitalized patients, RCT was performed by comparing the clinical outcome of 5-day (*n* = 200) and 10-day (*n* = 197) remdesivir treatments, with the 10-day group having worse clinical outcomes (*p* = 0.02). However, after adjusting for the severe patient bias in the 10-day group, the clinical status distributions at day 14 were similar between the two groups (*p* = 0.14). Although there was no placebo control, this implies that longer treatment periods did not improve the clinical condition [101]. The remdesivir treatment appears to have clinical benefits in patients with severe COVID-19 in several RCTs, but it is still unclear in patients with moderate COVID-19. There was no statistically significant difference in mortality when comparing the standard care group (*n* = 200), the remdesivir 5-day treatment group (*n* = 199), and the 10-day treatment group (*n* = 200). Additionally, there was no difference in clinical status at day 11 between the standard care group and the 10-day group. Although the clinical significance is uncertain, only the 5-day group showed a statistical improvement in the 11th day clinical status compared to the standard care group [102]. For the above reasons, on 19 November 2020, the FDA approved emergency use of remdesivir in hospitalized patients with severe COVID-19 [43]. However, the efficacy and optimal duration of treatment (5-day versus 10-day treatment) for moderate COVID-19 patients are controversial and require more research.

#### 5.1.2. Chloroquine and Hydroxychloroquine

Chloroquine, which is used for malaria or autoimmune diseases, has already shown antiviral effects on SARS-CoV-2. This drug raises endosomal pH by being passively diffused to endosomes and protonated, which induces interruption of viral fusion and entry into cells [103]. Chloroquine and its less toxic derivative, hydroxychloroquine, are known to inhibit SARS-CoV-2 proliferation through various pathways (Figure 6). Chloroquine also blocks the terminal glycosylation of ACE2 receptors. Additionally, elevated pH interrupts activation of cathepsin which cleaves the spike protein into S1 and S2 subunits. Furthermore, MAP-kinase which is needed for assembly of SARS-CoV-2 is downregulated by chloroquine [104]. Another study indicated that chloroquine has anti-inflammatory effects by interfering with the toll-like receptor and type I interferon signaling pathways [105]. Additionally, both chloroquine and hydroxychloroquine effectively inhibited SARS-CoV-2 reproduction in vitro [104,106]. Nonetheless, meta-analyses showed that chloroquine usage for treatment of COVID-19 patients is still controversial. Some studies consider chloroquine or hydroxychloroquine to be a promising treatment for COVID-19 [107,108]. In contrast, several studies indicate that benefits could not be found from the treatments and even that mortality increased with the use of chloroquine [109,110,111]. In addition, several recent RCTs have not found clinical benefits of chloroquine or hydroxychloroquine treatment [112,113,114]. The 28-day mortality rates in a RCT comparing a hydroxychloroquine group (*n* = 1561) and a standard care group (*n* = 3155) in hospitalized COVID-19 patients were 27.0% in the hydroxychloroquine group and 25.0% in the standard care group. In addition, the hydroxychloroquine group had a lower discharge rate from hospital (59.6% versus 62.9%) [112]. Another RCT, comparing a standard care group with hydroxychloroquine (*n* = 97) to a standard care group without hydroxychloroquine (*n* = 97), yielded similar results. Treatment of hydroxychloroquine could not lower the 28-day mortality (6.2% in the hydroxychloroquine group and 5.2% in the standard group, *p* = 0.77) and the need for a mechanical ventilator (4.1% in the hydroxychloroquine group and 5.2% in the standard care group, *p* = 0.75) [113]. Additionally, treatment of non-hospitalized mild COVID-19 patients with hydroxychloroquine could not statistically reduce the risk of worsening symptoms. On day 14, 24% of the hydroxychloroquine group had persistent symptoms, while 30% of the placebo group had persistent symptoms (*p* = 0.21) [114]. Considering the above RCT results, the FDA did not recommend the use of chloroquine or hydroxychloroquine, because no distinct clinical benefits (reduced mortality or shorter recovery time) were identified [43].

Recent research has also focused on the preventive role of hydroxychloroquine, but clinical evidence is insufficient [115,116]. The RCT of 821 asymptomatic patients with a moderate or high risk of COVID-19 exposure randomized participants to receive hydroxychloroquine (*n* = 414) or placebo (*n* = 407). However, symptomatic illness between the hydroxychloroquine and the placebo group showed no significant difference (11.8% versus 14.3%, *p* = 0.35) [115]. Similarly, another RCT with 2314 healthy contacts (*n* = 1116 in the hydroxychloroquine group, *n* = 1198 in the general treatment group) also did not show that hydroxychloroquine treatment was associated with symptomatic COVID-19 infection or low transmission [116].

### 5.2. Glucocorticoids (Dexamethasone) as an Anti-Inflammatory Drug 

Glucocorticoids are the most common anti-inflammatory treatments and they inhibit the transcription of inflammatory mediators, which decreases the levels of IL-6 or TNF-α. Thus, glucocorticoids have been widely used to mitigate lung inflammation in severe MERS and SARS patients, even though it has been shown to delay viral clearance in SARS patients [117,118]. In addition, a meta-analysis reported that glucocorticoids therapy was associated with an increased risk of mortality and secondary infection in patients with influenza-induced pneumonia [119]. Glucocorticoids are being used with caution in patients with severe COVID-19, because side effects can occur. Recent studies have shown that dexamethasone treatment may be beneficial for patients with severe COVID-19 who need oxygen supplementation [120,121,122]. A meta-analysis on 851 patients of seven RCTs also reported that glucocorticoids therapy reduced all-cause mortality (risk ratio 0.75, *p* = 0.02) and mechanical ventilator duration (mean difference −4.93 days, *p* < 0.001) [120]. An RCT that included 6425 hospitalized patients concluded that the use of dexamethasone decreased the “28-day mortality” rate as an indication of short-term mortality compared to control patients who received respiratory support (29.3% vs 41.4%), but not in patients who did not receive respiratory support (17.8% vs 14.0%). However, this benefit of dexamethasone was observed only in patients who needed supplemental oxygen and not in patients who did not [121]. Similarly, the use of dexamethasone was beneficial for alleviating tissue injury in COVID-19 patients with ARDS. The dexamethasone group with standard care had a lower mean sequential organ failure assessment score (6.1) compared to the control group (7.5). In addition, the need for a ventilator was significantly reduced during the first 28 days in the dexamethasone group (22.4 vs 24 days) [122]. Only dexamethasone seems to be proven to decrease the mortality of severe COVID-19 patients. However, doses and formulations of glucocorticoids treatment vary among patients. Therefore, there are only small RCTs for other glucocorticoids, such as methylprednisolone and hydrocortisone. Thus, medical evidence for using methylprednisolone and hydrocortisone is trivial [123]. For example, when comparing the methylprednisolone treatment group (*n* = 194) versus the placebo group (*n* = 199) in an RCT of 393 hospitalized patients, there was no significant difference in the 28-day mortality. However, patients over the age of 60 had a lower mortality rate in the methylprednisolone group [124]. Another RCT for 21-day mortality of 149 patients with mechanical ventilation showed that the hydrocortisone group (32 of 76, 42.1%) was numerically lower than the placebo group (37 of 73, 50.7%), but this result was not statistically valid [125]. More follow-up studies seem to be needed to evaluate the efficacy and clinical benefits of methylprednisolone and hydrocortisone.

## 6. Potential Therapeutics for COVID-19 Treatments and Their Disadvantages

Unlike the drugs described in Section 5, the following drugs are still in clinical trials, so their efficacy has not yet been demonstrated. However, they have been used in other diseases and their stability has been verified to some extent. Here, their therapeutic effects and disadvantages are reviewed. Additionally, other potential therapeutics are being studied to determine if they have the potential to be developed for COVID-19 treatment (Summarized in Figure 7).

### 6.1. Immunosuppressants for Treatment of Arthritic Diseases 

The SARS-CoV-2 infection causes systemic inflammation and a resulting cytokine storm, which in turn leads to multi-organ system failure, the leading cause of death in COVID-19 patients. In particular, leukocytosis, lymphopenia, neutropenia, and a dramatic increase in serum inflammatory cytokines, which include TNF-α, IFN-γ, IL-1β, and IL-6, are typical symptoms seen in severe COVID-19 patients. Therefore, various anti-inflammatory agents that have been used against arthritic diseases are administered to COVID-19 patients to control systemic inflammation and prevent organ failure.

#### 6.1.1. Anakinra, a Recombinant IL-1 Receptor Antagonist Protein

Anakinra, a recombinant IL-1R antagonist protein, has been used to treat rheumatoid arthritis. It inhibits the signaling pathways through activated IL-1R by interrupting interaction of IL-1α and IL-1β with IL-1R [126]. Thus, anakinra was expected to benefit severe COVID-19 patients with systemic inflammation and ARDS. A recent retrospective cohort study found that high-dose anakinra treatment was safe for COVID-19 patients with ARDS, which led to clinical improvement in 72% of patients. Among them, 90% survived after high-dose anakinra treatment, while only 56% survived in the standard group taking primarily hydroxychloroquine. Bacteremia with bacteria in the bloodstream occurred in 14% of patients (3 out of 29) in the anakinra treatment group and 13% (2 of 16 patients) in the standard group [127]. Another prospective cohort study also reported that anakinra reduced the mortality rate of COVID-19 patients with severe pneumonia. Specifically, the anakinra group had lower ICU hospitalization or mortality rate (13 out of 13) than the group receiving only remdesivir treatment [128].

#### 6.1.2. Tocilizumab, a Recombinant Humanized Anti-IL-6 Receptor Antibody

Tocilizumab, which is a recombinant humanized anti-IL-6 receptor antibody, suppresses the IL-6-mediated inflammatory response. Tocilizumab is authorized for relieving autoimmune diseases, including rheumatoid arthritis and temporal arteritis. Particularly, proinflammatory cytokine IL-6 is considered to be a key cytokine that leads to systemic inflammation or cytokine storms [129]. Because COVID-19 patients have overactivated T-lymphocytes and macrophages and a high serum level of inflammatory cytokines, tocilizumab is considered to be a possible treatment [130]. Tocilizumab was administered to severe COVID-19 patients to alleviate inflammation and organ failure. In China, COVID-19 patients with respiratory failure were reported to have lower CRP levels and relieved cough and fever after treatment with tocilizumab [131]. Another study in 21 patients with severe COVID-19 found that treatment with tocilizumab lowered CRP levels and improved hypoxemia and fever [132]. In addition, a systematic analysis of 10 studies that included 352 COVID-19 patients with pneumonia found that tocilizumab treatment lowered CRP levels and relieved hyperinflammatory conditions [133]. Another systematic review found that the tocilizumab-treated groups (16.3%, 240 out of 39) had a numerically (not statistically) lower mortality rate than the control group (24.1%, 85 out of 352). Another systematic review found that the tocilizumab-treated groups (16.3%, 240 out of 39) also had a numerically (not statistically) lower mortality rate than the control groups (24.1%, 85 out of 352) [134]. Considering that the proportions of patients in the ICU were 35.1% in the tocilizumab treatment group and 15.8% in the control group, tocilizumab may offer possible benefits for COVID-19 patients. However, bowel perforations and serious secondary infections have been reported in patients taking tocilizumab for a long period of time [135,136]. It remains controversial whether the benefits of tocilizumab outweigh the side effects that include latent infections and perforation of the lower gastrointestinal tract.

#### 6.1.3. Baricitinib, a Small Molecule That Inhibits Selective Janus Kinase (JAK) 1 and 2

Baricitinib, a selective Janus kinase (JAK) 1 and 2 inhibitor, is used to relieve inflammation. As it was approved for the treatment of patients with rheumatoid arthritis, its efficacy and safety have been demonstrated to some extent [137]. Interestingly, it has antiviral effects on COVID-19, as well as anti-inflammatory effects. Baricitinib has a high affinity for AP2-associated protein kinase 1 (AAK1), which is needed for the clathrin-mediated endocytosis of SARS-CoV-2 [99]. Additionally, unlike other AAK1 inhibitors such as erlotinib or sunitinib, has high affinity, which can suppress AAK1 at therapeutic doses (2–4 mg/day) to minimize side effects [138]. Treatment of COVID-19 pneumonia patients with baricitinib at 4 mg/day significantly improved clinical parameters, including CRP, arterial oxygen saturation, and fever, compared to controls. In addition, the baricitinib group did not require ICU hospitalization and no adverse reactions were reported [139]. However, there are still concerns about the use of baricitinib. INF-γ is primarily activated by the JAK-STAT signaling pathway. INF-γ-mediated gene expression controls antiviral response and rapid virus clearance [140]. Therefore, it is recommended to treat patients with baricitinib for a short period (7–14 days) to avoid opportunistic viral infections due to prolonged use. Nonetheless, there are no clinical trials with a proper control group, so further study of baricitinib is needed.

### 6.2. Inhibition of Specific N-Linked Glycosylation in the S1 Protein

As explained in Section 2.4, glycosylation of the S1 receptor-binding domain is important for modulation of viral entry into host cells. When S1 is glycosylated, it undergoes a dramatic conformational rearrangement, which is essential for fusion between the virus and its host cell membrane [30]. Each monomer of S, which is evenly distributed on the virion surface as a trimer, has 22 N-linked glycosylation sites. Thus, blocking N-linked glycosylation of the spike protein could interrupt viral entry into host cells [141]. However, non-specific inhibition of glycosylation sites could induce an increase in the binding of SARS-CoV-2 to the ACE2 receptor. For example, glycosylation at N90 of the S1 protein interrupts binding to ACE2, but glycosylation at N322 stimulates binding to ACE2 [30]. Thus, non-specific inhibition of the N-linked glycosylation site may have the undesirable effect of enhancing the binding of SARS-CoV-2 to ACE2. Currently, several α-glucosidase inhibitors, such as miglitol and celgosivir, are attracting attention, but clinical trials are not ongoing [142].

### 6.3. Modulation of ACE2 Expression by ACE Inhibitors (ACEi) and Angiotensin II Receptor Blockers (ARB) 

SARS-CoV-2 infects host cells through ACE2, and COVID-19 patients show a disabled RAS system. Patients with hypertension may have been taking ACEi or ARB. These medications induce the overexpression of ACE2 via various pathways to modulate the RAS system. Several studies indicated that ACEi- or ARB-mediated ACE2 overexpression can increase viral load and make host cells more vulnerable to SARS-CoV-2 [143,144]. However, COVID-19 patients also show downregulation of ACE2 receptors and severe systemic inflammation. ACEi or ARB can inhibit inflammatory reactions. Thus, meta-analyses indicate that a history of taking ACEi or ARB did not increase patient mortality, and rather these medications reduced the need for ICU hospitalization and mechanical ventilation equipment [145,146]. In addition, major cardiac societies, such as the American Heart Association, the American Heart Association, and the European Heart Association, did not agree with the correlation between taking ACEi or ARB and a high mortality rate and also recommended patients not stop taking ACEi or ARB. Analysis of animal models also supports that treatment with ARBs such as losartan would be beneficial as they exert cardiopulmonary protection by blocking RAS cascades [147]. Whether taking ACEi or ARB is beneficial for COVID-19 patients is still controversial, so further research is needed to ensure the safety of using ACEi or ARB. ACE2 is a receptor for SARS-CoV-2, but it is also an important modulator of the RAS system. Therefore, blocking the binding of SARS-CoV-2 to ACE2 would be an ideal target rather than down- or upregulating ACE2. Currently, worldwide attempts to develop monoclonal antibodies that target the spike protein, especially its RBD region, have been reported. Additionally, they show promising results in vivo and in vitro [148].

### 6.4. Blocking of Spike Protein Cleavage by Inhibiting the Activity of TMPRSS2 

SARS-CoV-2 spike proteins have to be cleaved for cell entry. Furin cleaves the spike protein into S1 and S2, then TMPRSS2 cleaves S2’ sites. Spike protein cleavage by purine and TMPRSS2 is required for ACE2 recognition and cell entry. Inhibition of their proteolytic activity is a plausible strategy to block host cell infection. It is well known that camostat mesilate, a synthetic serine protease inhibitor, blocks various inflammatory proteases, such as TMPRSS2 and trypsin. TMPRSS2 is essential for cell entry of SARS-COV-2, but not for host homeostasis or development [149,150]. An in vitro experiment showed that camostat mesilate treatment significantly reduced the infection of SARS-CoV and HCoV-NL63 in HeLa cells [151]. Another study also indicates that infection by SARS-CoV-2 into Calu-3 lung cells was inhibited by camostat mesilate [28]. Meanwhile, camostat mesylate tablets have been approved in Japan as a treatment for chronic pancreatitis, providing some dosage and safety. Therefore, several phase 2 clinical RCTs using camostat mesilate for COVID-19 patients are underway [132]. In addition, clinical trials using nafamostat mesylate, a derivative used as a treatment for chronic pancreatitis and an inhibitor of TMPRSS2, are being conducted worldwide. Furthermore, furin was once considered to be a key enzyme for viral entry and activation. A study reported that furin and its cleavage site were essential for cell-to-cell fusion or entry into lung cells [152]. However, more recent studies suggest that only furin inhibitors do not prevent viral infection and spread, although purin activation may promote infection [153,154]. Thus, inhibitors of TMPRSS2 or of both TMPRSS2 and furin would be ideal targets. Meanwhile, arbidol, which is used to treat the influenza virus, inhibits the trimerization of the spike glycoprotein, which is key for host cell adhesion and hijacking, indicating the potential of arbidol to treat COVID-19 [96].

### 6.5. Inhibitors of C3 and C5 Proteins in the Complement System

The complement system is also one of important components of the innate immune response. Thus, severe inflammation in COVID-19 patients appears to be due to overactivation of the complement-related system. An immunochemistry study indicated high expression levels of mannose-binding lectin, C4, C3, and the membrane attack complex in lung cells of COVID-19 patients [155]. Another study also revealed that SARS-CoV infection increased the products of activated C3. Additionally, C3-deficient mice had significantly lower levels of lung damage and inflammation regardless of viral load [156]. Additionally, accumulation of terminal complement components and overactivation of mannose-binding, lectin-associated serine protease (MASP) 2 have been observed in severe COVID-19 patients. In particular, the spike protein is deposited along with C5-9 and C4d in microvessels, which causes tissue damage across multiple organs [157]. Therefore, several clinical trials are underway that use complement inhibitors to relieve inflammation and ARDS. A humanized monoclonal antibody against C5 (eculizumab) is in clinical trials for patients with ALI or ARDS. Additionally, the C3 inhibitors (apellis and AMY-101) and the MASP blocker (narsoplimab) are in clinical trials as immunomodulators [158]. Since the complement system appears to be associated with symptom severity in COVID-19 patients, more studies and RCTs are needed to demonstrate the safety and efficacy of these drugs.

### 6.6. Neutrophil Extracellular Trap (NET)-Associated Therapeutics

In response to infection, neutrophils move to the inflammatory site. The pattern recogniiton receptor (PRR) of neutrophils bind to the pathogen-associated molecular pattern (PAMP), activating the antimicrobial signaling cascade, which triggers the release of neutrophil extracellular traps (NETs), oxidative burst, and phagocytosis. Thus, infections induce a unique form of cell death called NETosis, in which neutrophils release decondensed chromatin and antimicrobial maromolecules into extracellular space [159]. Similarly, several studies have reported that SARS-CoV-2 infection induces NET formation and NETosis [160,161]. The detailed mechanism for NET formation is not clear, although the peptidylarginine deiminase (PAD)4 enzyme, which is stimulated by ROS, unwinds the core histones from DNA. Additionally, myeloperoxidase migrates to the nucleus and decondensates chromatin [159]. NET formation is an immune response to infection, but under certain conditions, it could cause excessive inflammatory reactions, hypercoagulable state, and tissue damage. NETs break down coagulation inhibitors and stimulate platelet aggregation, as NETs act as ligands for Toll-like platelet receptors [162]. Therefore, intravascular NET activity appears to exacerbate the lung injury and thrombosis in COVID-19 patients [160,161,163,164]. The serum levels of NET formation markers (myeloperoxidase and citrullinated histone H3) were elevated in COVID-19 patients in one study [160]. In addition, COVID-19 patients with neutrophilia tended to have more frequent thromobotic events and poor prognosis [161,163]. Additionally, lung autopsy of COVID-19 patients revealed NETs containing microthrombi, while the factors that triggered NETs formation were significantly increased [164]. Therefore, antiplatelet compounds (cicaprost), neutrophil elastase inhibitors (sivelestat), and PAD inhibitors (hydrochloride) could be considered possible treatment options that could alleviate thrombosis and lung injury of COVID-19 pateints [40].

### 6.7. Ivermectin

Ivermectin is an antiparasitic drug used to treat onchocerciasis, scabies, and helminthiases. In addition, ivermectin is a multifunctional drug that shows antibacterial, anticancer, and anti-inflammatory effects [165]. Recently, ivermectin has attracted attention for its antiviral effects. Importin α/β-1 nuclear transport protein is required for intranuclear trafficking of proteins of several RNA viruses. Ivermectin binds to the importin α/β-1 nuclear transport protein, preventing the intranuclear transport of SARS-CoV-2 [166]. In a 48-h cell culture experiment, treatment with ivermectin 2 h after SARS-CoV-2 infection resulted in a ~5000-fold reduction in viral RNA [167]. However, the plasma concentration required for antiviral efficacy detected in vitro requires a dose 100 times higher than that approved for human use [168]. In a retrospective analysis comparing patients with COVID-19 who received one or more doses of ivermectin (*n* = 173) with those who received conventional treatment (*n* = 103), all-cause mortality was lower in the ivermectin group than the conventional treatment group (odds ratio 0.27, *p* = 0.03). The mortality benefits have been shown to be limited to severe patient subgroups. However, the therapeutic interventions were not standardized and are unclear. Furthermore, no virological evaluation has been performed to confirm the effectiveness of ivermectin treatment [169]. Therefore, the efficacy of ivermectin in COVID-19 patients is unpredictable and more clinical follow-up is needed. 

## 7. Conclusions

At present, as there is no licensed treatment for COVID-19, only a combination of antiviral and anti-inflammatory drug treatments are used. Remdesivir is a representative antiviral treatment that interrupts viral replication. Anti-inflammatory drugs such as glucocorticoids are used for severe or critical patients. Current CDC guidelines do not recommend the use of most test drugs, except remdesivir and dexamethasone under certain conditions. However, recombinant therapeutic proteins such as anakinra and tocilizumab, as well as a small molecule that targets JAK, baricitinib, are undergoing clinical trials to test their anti-inflammatory effects. Additionally, the serine protease inhibitor camostat-mesilate, which inhibits TMPRSS2 activity, as well as several α-glucosidase inhibitors, such as miglitol or celgosivir, could inhibit virus attachment to the ACE2 receptor. The modulators of ACE2 expression, ACEi and ARB, also inhibit virus attachment. Inhibitors of complement C3 and C5 activation could inhibit inflammation. However, not much is known about treating COVID-19 by inhibiting virus infection and growth. Therefore, it is necessary to share information across clinical trials and research studies into various drugs that are in progress in order to develop potential therapeutics for COVID-19. This review intends to provide new insights into the development of therapeutics by understanding the various clinical and basic research studies currently underway.

## Figures and Tables

**Figure 1 ijms-22-01737-f001:**
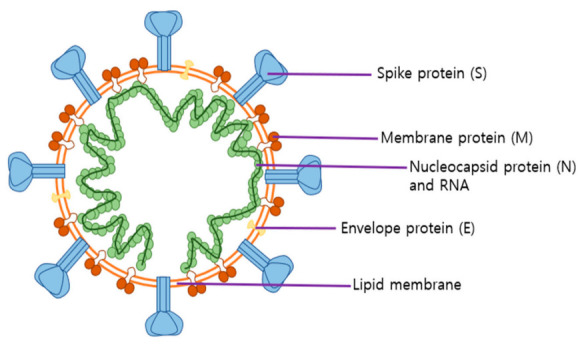
Roles of severe acute respiratory syndrome coronavirus 2 (SARS-CoV-2) structural proteins. The spike protein (S), which provides the crown shape, mediates the binding of the virus to the host’s angiotensin-converting enzyme 2 (ACE2) receptors and determines its affinity for the host. The nucleocapsid protein (N), which encloses the viral genome, is known to regulate transcription and immunoreaction. The membrane protein (M), which is the most common structural protein, seems to be involved in viral assembly and maturation. The envelope protein (E), which is the smallest structural protein, functions as an ion channel and assists in viral assembly and budding.

**Figure 2 ijms-22-01737-f002:**
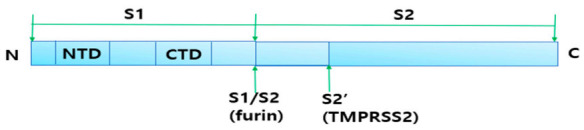
Spike (S) protein cleavage sites. The spike protein (S) trimer is a transmembrane protein, which gives the virus its crown-like appearance. The spike protein with 22 glycosylation sites has two ectodomains, S1 and S2, which are mandatory for recognition and binding to the host’s angiotensin-converting enzyme 2 (ACE2) receptors and are involved in viral fusion, respectively. The S1 domain contains 2 subdomains that include an N-terminal domain (NTD) and a C-terminal domain (CTD), which both could work as receptor-binding domains (RBD). The spike protein needs to be cleaved by the host enzymes furin and transmembrane protease serine subtype (TMPRSS2). The S1/S2 site is cleaved by furin and the S2’ site is cleaved by TMPRSS2. These proteases are essential for receptor recognition and viral entry into host cells.

**Figure 3 ijms-22-01737-f003:**
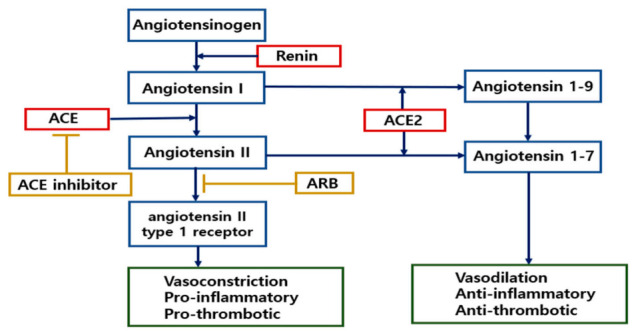
Role of ACE and ACE2 in modulating blood pressure in the renin–angiotensin system (RAS). The renin-angiotensin system (RAS) is an essential hormone system that regulates blood pressure and fluid balance in the body. Renin converts angiotensin to angiotensin I and then ACE converts angiotensin I to angiotensin II. Angiotensin stimulates production of cytokines and exerts proinflammatory, prothrombotic, and vasoconstrictive effects. ACE2 converts angiotensin II to angiotensin 1-7 and exerts opposing effects. Most common antihypertensive drugs are ACE inhibitors and angiotensin II receptor blockers (ARBs) and lower blood pressure by blocking angiotensin II related signaling pathways.

**Figure 4 ijms-22-01737-f004:**
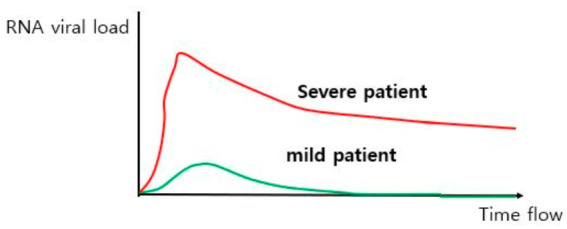
Difference in viral load between mild and severe COVID-19 patients. The viral load seen in COVID-19 patients was positively correlated with symptom severity, greater mortality, and longer recovery time. In particular, viral load in severe or critical patients showed two features compared to mild patients: higher peak viral load and delayed viral clearance.

**Figure 5 ijms-22-01737-f005:**
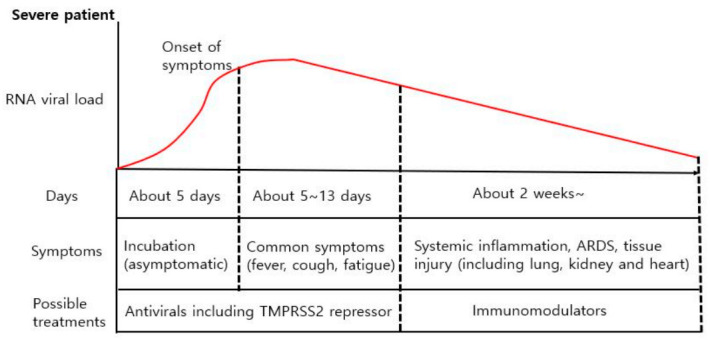
Possible treatment strategies for progressive symptoms in patients with severe COVID-19. The incubation period for COVID-19 patients is about 5~6 days (but could be up to 14 days). After the incubation period, patients have common and relatively mild symptoms, including fever and cough. About 80% of COVID-19 patients recover spontaneously, but about 20% develop serious symptoms such as increased inflammation markers and pneumonia. Persistent inflammation and delayed virus removal can cause serious tissue damage to the lungs, kidneys, and heart, which can be fatal. Therefore, immunomodulators should be administered to severely ill patients.

**Figure 6 ijms-22-01737-f006:**
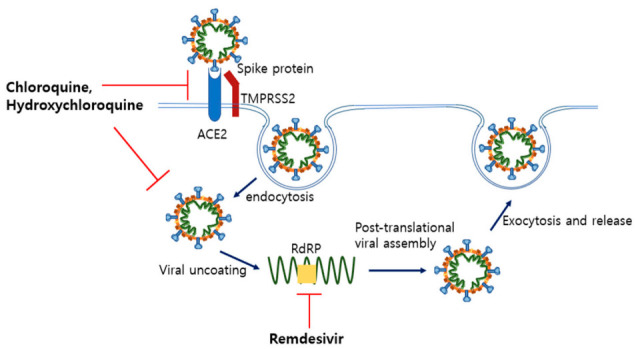
Mechanism of antiviral drugs inhibition of SARS-CoV-2. Chloroquine and hydroxychloroquine raise endosomal pH and block viral uncoating. Additionally, they seem to interrupt glycosylation of the spike protein, which is expected to block SARS-CoV-2 recognition of ACE2. However, recent clinical trials did not show any benefits, i.e., decreased mortality or short recovery time. Thus, the FDA recommended clinicians not use chloroquine or hydroxychloroquine for patients. Remdesivir is a nucleotide analogue that blocks the activity of RNA-dependent RNA polymerase (RdRP). Additionally, clinical trials showed that the use of remdesivir shortened the recovery time of COVID-19 patients.

**Figure 7 ijms-22-01737-f007:**
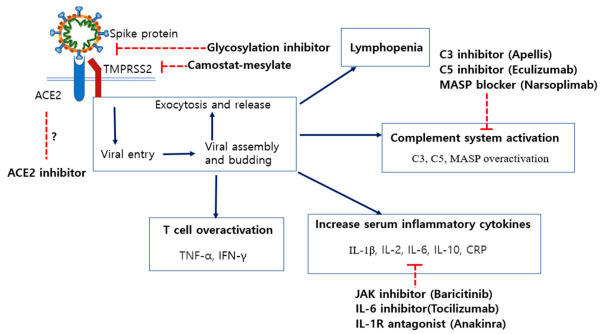
Possible treatment strategies for COVID-19 patients that block receptor recognition or inhibit immune reactions. Spike protein cleavage is essential for ACE2 recognition. The synthetic serine protease inhibitor, camostat–mesilate, seems to block viral entry by inhibiting the activity of TMPRSS2. Additionally, several glycosylation sites of the spike protein could be targets to lower the affinity of the spike protein for ACE2. Thus, specific glycosylation inhibitors could be an option for COVID-19 treatment. Severe inflammation causes pneumonia, critical tissue injury, or even cytokine storms in COVID-19 patients. Therefore, proven immunomodulators, including the JAK inhibitor baricitinib, the IL-6 inhibitor tocilizumab, or the IL-1R antagonist anakinra, could be beneficial treatments. Additionally, complement system inhibitors are expected to alleviate inflammation and overactivation of immune cells. Several clinical trials are ongoing and further study is needed to ensure their efficacy and safety.

## Data Availability

No new data were created or analyzed in this study. Data sharing is not applicable to this article.

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
