# Peer review of "Understanding Viral Infection Mechanisms and Patient Symptoms for the Development of COVID-19 Therapeutics"

_ijms, 2021, doi:10.3390/ijms22041737_

Round 1

Reviewer 1 Report

The manuscript presents a review of some aspects of SARS-CoV-2 infection and COVID-19 therapeutics. The manuscript is well written and structured, but does not add much to a subject that has been extensively reviewed in the last months.  Some sections lack information or are oversimplified, for example in section 4.1 no reference is given for the study mentioned in lines  271-274 and in line 277 it is not explained that the correlation with clinical severity found in reference 64 was not statistically significant. Things are moving very fast and section 5 is already outdated.

Author Response

Dear Editor-in Chief,

Thank you for allowing us to revise our paper. The constructive advice of the peer reviewers has substantially improved our paper. Attached are our detailed point-by-point responses to their comments. We apologize for the late revision and would like to thank you for your patience.

All changes in the manuscript have been marked with red colour.
On the behalf of the authors
Yours sincerely,

Kyoung-Soo Kim

Responses to the comments

Reveiwer1

The manuscript presents a review of some aspects of SARS-CoV-2 infection and COVID-19 therapeutics. The manuscript is well written and structured, but does not add much to a subject that has been extensively reviewed in the last months.

1) Some sections lack information or are oversimplified, for example in section 4.1 no reference is given for the study mentioned in lines 271-274 and in line 277 it is not explained that the correlation with clinical severity found in reference 64 was not statistically significant.

Response 1) We put the references to the lines that the reviewer pointed out (line 332, reference [86]). Also, explanation for reference [87] was added in line 334-335. Thank you.

2) Things are moving very fast and section 5 is already outdated.

Response 2) In Section 5, we added more recent clinical data and research trends about remdesivir, hydroxychloroquine, and dexamethasone.

Reviewer 2 Report

The ongoing pandemic of coronavirus disease 2019 (COVID 19) has led to 46.2 million infections and almost 1.2 million deaths worldwide up until now. With hundreds of vaccine candidates under clinical trials, the odds of more than one vaccine getting approved for vaccination are better. However, it is equally important to find avenues like antiviral therapeutics, drugs to limit viral pathogenesis and clinical outcomes, and viral transmission control measures in the event of questionable efficacy of marketed vaccines in the wider population. The current review from Choi et al tries to discuss the current understanding of the SARS-CoV-2 mechanism of infection, pathogenesis of COVID 19, major clinical findings, treatment strategies, and current and prospective therapeutic options for the treatment of COVID-19. I found some sections of the review very well written with a sound review of present literature and discussing the current state of knowledge in the field. But overall, the review seems likely to be written hastily with very poor editing and needs a lot of corrections. My comments are as follows;

  1. Drug repositioning is mentioned in the abstract to discuss the potential use of current antivirals and anti-inflammatory agents in COVID treatment. Authors should discuss the concept of drug repositioning, tools used and challenges to give a brief introduction about the topic.
  2. There is a discrepancy in overall death-to-case rate in abstract (3.3%) and introduction (3.8%) while the actual number which is around 2.6%. Authors should correct this.
  3. In section 2.2, while discussing the roles of SARS-CoV-2 structural protein for infection, authors could use recent research findings illustrating mutation in spike protein and their effect on viral transmission and pathogenesis potential.
  4. Section 2.3: line 101 states ‘transmitted aerosol carrying virus particles spreads to the respiratory tracts. I think the general pattern of SARS-CoV-2 transmission is the inhalation of virus-containing droplets or direct mucosal contact with virus-contaminated surfaces.
  5. Does SARS-CoV-2 spike protein affinity to ACE2 can be a measure of pathogenesis? Because spike protein of NL63 lineage of coronavirus has a similar affinity to ACE2 as SARS-CoV-2 but is drastically less pathogenic.
  6. Provide references for line 145 linking interruption of host cell signaling pathways leading to ARDS during COVID-19.
  7. Line 176 claims that COVID-19 infection can lead to an increased risk of cardiac injury, myocarditis, and diabetes mellitus. Authors should discuss this in detail whether the presence of other confounding factors such as ACEI, ARB, or immunomodulators can affect the outcomes in patients.
  8. Authors should also discuss vascular damages in the lung leading to blood clotting in many COVID-19 patients while discussing COVID-19 pathogenesis in section 3. Authors should also revise section 3.1, many of the events leading to pathogenesis are discussed twice in the text.
  9. In subsection heading 5.1, Remdesivir and hydroxychloroquine are referred to as anti-inflammatory drugs. They are not. They have antiviral activity. Authors should correct this.
  10. In many instances, authors, while writing conclusive statements at the end of many sections, have generalized the findings in the field that lack conclusive evidence or are still being investigated. Authors should not state them as facts as most of them are still being heavily researched.
  11. The draft needs to undergo extensive modification and editing for writing style and grammar. There are discrepancies in text size in various sections.

Author Response

Dear Editor-in Chief,

Thank you for allowing us to revise our paper. The constructive advice of the peer reviewers has substantially improved our paper. Attached are our detailed point-by-point responses to their comments. We apologize for the late revision and would like to thank you for your patience.

All changes in the manuscript have been marked with red colour.

On the behalf of the authors

Yours sincerely,

Kyoung-Soo Kim

Reveiwer2

The ongoing pandemic of coronavirus disease 2019 (COVID 19) has led to 46.2 million infections and almost 1.2 million deaths worldwide up until now. With hundreds of vaccine candidates under clinical trials, the odds of more than one vaccine getting approved for vaccination are better. However, it is equally important to find avenues like antiviral therapeutics, drugs to limit viral pathogenesis and clinical outcomes, and viral transmission control measures in the event of questionable efficacy of marketed vaccines in the wider population. The current review from Choi et al tries to discuss the current understanding of the SARS-CoV-2 mechanism of infection, pathogenesis of COVID 19, major clinical findings, treatment strategies, and current and prospective therapeutic options for the treatment of COVID-19. I found some sections of the review very well written with a sound review of present literature and discussing the current state of knowledge in the field. But overall, the review seems likely to be written hastily with very poor editing and needs a lot of corrections. My comments are as follows;

  1. Drug repositioning is mentioned in the abstract to discuss the potential use of current antivirals and anti-inflammatory agents in COVID treatment. Authors should discuss the concept of drug repositioning, tools used and challenges to give a brief introduction about the topic.

Response1) We added the explanation of drug repositioning in line 47-55

  1. There is a discrepancy in overall death-to-case rate in abstract (3.3%) and introduction (3.8%) while the actual number which is around 2.6%. Authors should correct this.

Response 2) We have corrected it based on data released on December 29, 2020 (line 16, 37). Thank you.

  1. In section 2.2, while discussing the roles of SARS-CoV-2 structural protein for infection, authors could use recent research findings illustrating mutation in spike protein and their effect on viral transmission and pathogenesis potential.

Response 3) We have added the explanation of Spike protein mutations associated with increased viral transmissibility. For example, the D614G mutation has explained how the mutation increased its transmissibility. Also, several mutations in the S protein are described in lines 84-96. Thank you for your feedback.

  1. Section 2.3: line 101 states ‘transmitted aerosol carrying virus particles spreads to the respiratory tracts. I think the general pattern of SARS-CoV-2 transmission is the inhalation of virus-containing droplets or direct mucosal contact with virus-contaminated surfaces.

Response 4) In line 116-117, we describe more details that SAR-CoV-2 was transmitted by not only Aerosol but also inhaled droplet.

  1. Does SARS-CoV-2 spike protein affinity to ACE2 can be a measure of pathogenesis? Because spike protein of NL63 lineage of coronavirus has a similar affinity to ACE2 as SARS-CoV-2 but is drastically less pathogenic.

Response 5) NL63 lineage of coronavirus has a similar affinity but not that pathogenic, because it usually infects upper respiratory tract but not lower respiratory tract. So, it does not induce systemic disease. However, SARS-CoV is able to infect lower respiratory tract because it is able to grow in 37°C. Thus, SARS-CoV may induce pneumonia, ARDS and systemic diseases. If the Spike protein of SARS-CoV-2 has a higher affinity for ACE2, the cells expressing ACE2 become more susceptible, which may be more pathogenic.

According to the critical comments, we described more details in 92-95, and 260-263. Thank you for the comments.

  1. Provide references for line 145 linking interruption of host cell signaling pathways leading to ARDS during COVID-19.

Response 6) We added the reference [31] in line 159.

  1. Line 176 claims that COVID-19 infection can lead to an increased risk of cardiac injury, myocarditis, and diabetes mellitus. Authors should discuss this in detail whether the presence of other confounding factors such as ACEI, ARB, or immunomodulators can affect the outcomes in patients.

Response 7) We added explanations in line 188-198

  1. Authors should also discuss vascular damages in the lung leading to blood clotting in many COVID-19 patients while discussing COVID-19 pathogenesis in section 3. Authors should also revise section 3.1, many of the events leading to pathogenesis are discussed twice in the text.

Response 8) In Section 3.1, we removed the repeated explanation about pathogenesis that the reviewer pointed out. Also, we added section 3.3., explaining that COVID-19 induced tissue damage and clinical presentations in line 294-324. In particular, COVID-19 induced hypercoagulable state was mainly explained in line 299-307.

  1. In subsection heading 5.1, Remdesivir and hydroxychloroquine are referred to as anti-inflammatory drugs. They are not. They have antiviral activity. Authors should correct this.

Response 9) We corrected ‘anti-inflammatory’ to ‘anti-viral’ according to the comment. Thank you.

  1. In many instances, authors, while writing conclusive statements at the end of many sections, have generalized the findings in the field that lack conclusive evidence or are still being investigated. Authors should not state them as facts as most of them are still being heavily researched.

Response 10) We corrected the manuscript in section 4 and 5 by adding more recent reports and evidence. Thank you for the critical comments.

  1. The draft needs to undergo extensive modification and editing for writing style and grammar. There are discrepancies in text size in various sections.

Response 11) We corrected and edited it again with the help of English-editing company.

Round 2

Reviewer 2 Report

The review looks very comprehensive and can be accepted for publication.

Author Response

Dear Editor-in Chief,

Thank you for reading the entire manuscript carefully. The constructive advice has substantially improved our paper. Attached are our detailed point-by-point responses to the comments.

All changes in the manuscript have been marked with blue colour.

On the behalf of the authors

Yours sincerely,

Kyoung-Soo Kim

Response to reviewer’s comments

  1. lines 42/45. The statements about vaccines anti-Sars-Cov-2 are incorrect based on recent articles showing efficacy of different vaccines.

Response) We changed incorrect statements.

  1. lines 308/314. There are excellent reports describing in details the different myocardial involvements in Covid-19.

Response) We changed several wordings on myocardial involvements.

  1. There is increasing evidence of neutrophils and NETs involvement in Covid-19. These observations have therapeutic implications which have been omitted by the authors.

Response) We added section 6.6. NET associated therapeutics. Thank you for the comment.

  1. Points 5.1 and 5.1.2. It is unclear why chloroquine/hydroxychloroquine have been discussed in two separate points.

Response) We rearranged section 5.1.

  1. Point 5.2. Please, modify "Corticosteroids" to "Glucocorticoids" throughout the text.

Response) We corrected as pointed out.

  1. lines 565/569. There are at least a couple of papers showing that the administration of Tocilizumab can increase secondary infections.

Response) We added some references on side effects of tocilizumab.

  1. lines 690/691. This Reviewer has the impression that the authors have excessive enthusiasm for test drugs.

Response) We added that current CDC guidelines in line 705-706
